# Enhancing Cranio-Maxillofacial Fracture Care in Low- and Middle-Income Countries: A Systematic Review

**DOI:** 10.3390/jcm13082437

**Published:** 2024-04-22

**Authors:** Christian Deininger, Florian Wichlas, Marco Necchi, Amelie Deluca, Susanne Deininger, Klemens Trieb, Herbert Tempfer, Lukas Kriechbaumer, Andreas Traweger

**Affiliations:** 1University Clinic for Orthopedics and Traumatology, Paracelsus Medical University, Müllner Hauptstrasse 48, 5020 Salzburg, Austria; f.wichlas@salk.at (F.W.); k.trieb@salk.at (K.T.); l.kriechbaumer@salk.at (L.K.); 2Department of Surgery and Orthopaedics, Hospital Sterzing, Margarethenstraße 24, 39049 Sterzing, Italy; marco.necchi@hotmail.it; 3Institute of Tendon and Bone Regeneration, Spinal Cord Injury & Tissue Regeneration Center Salzburg, 5020 Salzburg, Austria; dr.ameliedeluca@gmail.com (A.D.); herbert.tempfer@pmu.ac.at (H.T.); andreas.traweger@pmu.ac.at (A.T.); 4Department of Urology and Andrology, Paracelsus Medical University, Müllner Hauptstrasse 48, 5020 Salzburg, Austria; s.deininger@salk.at; 5Department for Orthopaedics and Traumatology, Center for Clinical Medicine, Faculty of Medicine and Dentistry, Danube Private University, 3500 Krems, Austria

**Keywords:** cranio-maxillofacial injuries, low- and middle-income countries, education, patient transfer, off-label solutions

## Abstract

**Background**: Cranio-maxillofacial (CMF) injuries represent a significant challenge in low- and middle-income countries (LMICs), exacerbated by inadequate infrastructure, resources, and training. This systematic review aims to evaluate the current strategies and solutions proposed in the literature to improve CMF fracture care in LMICs, focusing on education, patient transfer, and off-label solutions. **Methods**: A comprehensive literature search was conducted using PubMed/Medline from January 2000 to June 2023. Studies were selected based on the Preferred Reporting Items for Systematic Review and Meta-analysis Statement (PRISMA). Solutions were categorized into three main areas: education (digital and on-site teaching, fellowships abroad), patient transfer to specialized clinics, and off-label/non-operative solutions. **Results**: Twenty-three articles were included in the review, revealing a consensus on the necessity for enhanced education and training for local surgeons as the cornerstone for sustainable improvements in CMF care in LMICs. Digital platforms and on-site teaching were identified as key methods for delivering educational content. Furthermore, patient transfer to specialized national clinics and innovative off-label techniques were discussed as immediate solutions to provide quality care despite resource constraints. **Conclusions**: Effective CMF fracture care in LMICs requires a multifaceted approach, prioritizing the education and training of local healthcare professionals, facilitated patient transfer to specialized centers, and the adoption of off-label solutions to leverage available resources. Collaborative efforts between international organizations, local healthcare providers, and educational institutions are essential to implement these solutions effectively and improve patient outcomes in LMICs.

## 1. Introduction

Accidents remain the leading cause globally of death and disability among the working population. Approximately 40 million people are temporarily injured, and another 100 million people are permanently impaired per year. Overall, between 80% and 90% of all accidents with significant injuries occur in low- and middle-income countries (LMICs), which currently comprise 137 countries [1,2]. A lack of resources results in non- or under-provision of care for injuries requiring surgical treatment, including a lack of specialized surgical staff, surgical equipment, and infrastructure [3]. Furthermore, rural areas are often difficult to access [4], and most deaths occur during the pre-hospital phase [5,6].

The combination of all these conditions leads to a large number of avoidable deaths of young patients and to a preventable loss of manpower, which affects the entire population [1]. As a solution, the UN has set up a series of Sustainable Development Goals for 2030, which includes the objective of ensuring universal access to surgery, including oral and craniomaxillofacial surgery for all individuals [7].

To improve the long-term medical situation, it is essential to adequately train a greater number of local personnel. Even though resources in LMICs are generally not up to the standard of HICs (high-income countries), they can lead to a significant improvement in local patient care if used correctly [8,9]. The aim should therefore be to promote the teaching of diagnostic and therapeutic measures to the local population [10,11,12,13]. The following systematic analysis will assess the current situation, utilizing a pars pro toto example.

CMFs (craniomaxillofacial fractures) usually occur in LMICs due to traffic accidents, where passive safety mechanisms such as seatbelts and helmets are lacking, and due to poor traffic infrastructure with a lack of lighting on interurban roads at night [14,15,16]. Due to the absence of CT scanners, a detailed description and classification of CMFs in LMICs are difficult to impossible. They typically result from high-velocity traumas, often manifesting as complex comminuted fractures of the midface [15,16].

Untreated facial trauma can lead to various consequences, including post traumatic deformity and disfigurement, functional impairment, infection, pain and discomfort, or complications in adjacent structures [17,18,19]. For example, a blow-out fracture of the orbital floor and a consequent subsidence of the affected eye may be followed by diplopia [20]. Swelling of the soft tissues or severe dislocation may cause upper airway obstruction, and a dislocated fracture through the upper dental row may disrupt occlusion and as a result hamper food intake and speech [21,22]. In addition, a connection between the oral cavity and the nasal cavity may develop due to the broken hard palate. If left untreated, this allows food to enter the nasal cavity during eating and drinking, leading to chronic choking [23]. If these fractures heal in this malposition, persistent impairments of the patient are to be expected [24]. Also, mandibular fractures constitute a significant portion of facial injuries, with Open Reduction Internal Fixation (ORIFs) being necessary in the majority of cases, excluding fractures involving the temporomandibular joint area, while post-treatment malocclusion and restricted mouth opening represent primary complications. The treatment is also hampered due to the lack of resources in LMICs [25,26,27].

In HICs, an accurate diagnosis of these fractures involves both a comprehensive physical examination and a thin-slice CT scan. This combination enables precise planning of the approaches and miniplate osteosyntheses employed [28,29,30,31]. Nevertheless, alternative treatment modalities for these fractures exist. This includes non-surgical options such as employing a Barton bandage—a head and mandible wrap ensuring a six-week rigid occlusion for dentition reduction—or treating fractures with Kirschner wires [32,33,34,35].

This systemic review discusses the challenges of CMF trauma in LMICs. This includes the necessity for a comprehensive approach that integrates education and training for local surgeons, access to digital learning platforms, and the adoption of off-label solutions. Empowering local surgeons with skills and knowledge is crucial for establishing sustainable surgical care in these regions. Effective implementation of solutions relies on collaboration between international organizations, local healthcare providers, and educational institutions, thus leading to significant improvements in patient outcomes in LMICs.

## 2. Material and Methods

### 2.1. Literature Search

From April to June 2023, a systematic literature search was conducted using PubMed/Medline to identify available studies on solution ideas to improve fracture care in the field of CMFs in LMICs. Only publications from the last 23 years were included (January 2000–June 2023). The search was performed using the following search terms in different combinations: LMIC, Africa, maxillofacial trauma, maxillofacial fracture, maxillofacial surgery, global surgery, low resources, teaching. The following combinations have been searched: “craniomaxillofacial trauma + africa”, “craniomaxillofacial trauma third world”, “craniomaxillofacial trauma low resource”, “craniomaxillofacial fracture low resource”, “craniomaxillofacial fracture third world”, ““maxillofacial surgery” LMIC”, ““maxillofacial surgery” “low resource”, ““facial fracture” Africa”, ““global surgery” “maxillofacial surgery”, “maxillofacial surgery + teaching”. The reference lists of the studies identified were also used to obtain additional relevant literature.

### 2.2. Study Selection

The author performed the study selection following the recommendations of the Preferred Reporting Items for Systematic Review and Meta- analysis Statement (PRISMA) [36]. The study selection process is shown in Figure 1. A total of 76 articles were identified through data bank research. An additional 21 articles were retrieved from other sources, such as library reviews. After removing duplicates, 84 articles remained. Titles and abstracts of 74 articles were studied. A full-text assessment for eligibility was performed of 33 articles. Forty-one studies identified with the above combinations were excluded as they did not pertain to CMFs. In conclusion, a total number of 23 articles were included in this systematic review. Only publications written in the English language were incorporated in this study.

The studies revealed an overlap in solutions aimed at enhancing CMF fracture care in LMICs, and they were not prospectively randomized, precluding statistical evaluation. Consequently, a qualitative comparison of diverse solution paths was conducted by analyzing the proposed strategies based on the existing literature.

### 2.3. Risk of Bias [37]

An analysis according to Cochrane Collaboration’s tool for assessing the risk of bias was not possible, because the included studies were not prospectively randomized trials. Due to the lack of such studies, case reports or case series were also included in this systematic review. The risk of bias must therefore be considered high.

### 2.4. Data Extraction and Categorization

A review of publications and data extractions were performed by the author. Initially, a table was created which contained the following characteristics of the respective studies: Author, year of publication, PMID. Further, all included studies were screened either as full text or abstract and assigned to one of the following solution categories:Category I: Teaching.Digital teaching;On-site teaching;Fellowships abroad.Category II: Transfer of the patient to specialized national clinics.Category III: Off-label and non-operative solutions.

## 3. Results of the Literature Search by Category

### 3.1. Category 1: Digital Teaching

By far, the largest proportion of the studies found the education and training of the local staff to be lacking. In general, the projects described below do not focus on one of the defined categories, but rather focus on optimizing the teaching of surgeons in LMICs by linking various approaches. For example, some online teaching platforms offer fellowship programs or try on site teaching techniques [38].

#### 3.1.1. Digital Teaching Platforms

Several authors address the possibility of sharing surgical techniques for the care of CMFs online using digital platforms.

Acero, J. describes in 2021 the digital teaching offers of the International Association of Oral and Maxillofacial Surgeons (IAOMS), which pursues a global orientation of teaching for oral and maxillofacial surgeons. Another option is the “European Lecture Series” of the European Association for CMF surgery (EACMFS) [38,39,40]. Both websites show basic techniques as well as advanced surgical procedures. In addition to online lectures, surgical videos are available. As the authors describe, such online courses contribute to a significant increase in knowledge for oral and maxillofacial surgeons worldwide [38].

While the homepage “https://www.schoolforsurgeons.net/, accessed on 17 March 2024 of the Royal College of Surgeons in Ireland was developed specifically for the education of surgeons in LMICs, the “SCORE Project” of the Surgical Council on Resident Education was designed for the online education of residents in the USA. The second online teaching platform alone offers 254 modules for continuing education in oral and maxillofacial surgery.

A study by Goldstein et al. from 2014 aimed to evaluate the feasibility of Internet-based educational platforms for enhancing the education and training of surgical providers in LMICs. Seventy-five surgical faculty and trainees from 12 countries of East, Central, and Southern Africa were given access to online curricula for 60 days. Participants completed an anonymous online survey. Results showed that both curricula were well rated, with no significant differences in participants’ willingness to use and recommend either platform to colleagues [41,42,43].

A study by Derbew et al. from 2006 discusses the Ptolemy Project, a model of electronic access to medical literature specifically designed for surgeons in developing countries, particularly in East Africa. The project enables East African surgeons to become research affiliates of the University of Toronto, granting them access to the university library’s comprehensive resources through a secure system that monitors and evaluates their usage. By providing electronic resources and training opportunities, the project aims to empower African surgeons to contribute to advancements in surgical practice and healthcare in their respective regions [44,45].

Harris et al. examined the global reach of social media among oral and maxillofacial surgeons and characterized the profile activity and demographic characteristics of followers. The study focuses on a single oral and maxillofacial surgery-related Instagram account and analyses variables such as total number of followers, profile views, media content posts, likes, comments, saves, impressions, and reach. The study concludes that social media platforms have the potential to enhance global collaboration and facilitate the dissemination of surgical knowledge and expertise [46].

Ambroise et al. reported on their telemedicine project in their 2018 study. Initially, the infrastructure and the team were set up to support colleagues from Mali about the therapy of complex cases in oral and maxillofacial surgery. Primary review of imaging (X-rays and computed tomography scans) and review of patient history were conducted. Therapy was carried out either by the local surgical team or in collaboration with the European colleagues during a humanitarian mission. Real-time exchange of expertise with Malian colleagues, shared therapeutic decision making, academic value, and anticipation of anesthesia and surgical needs prior to surgery [47].

#### 3.1.2. On-Site Teaching

Mohan’s paper from 2018 discusses the engagement of surgical trainees in global surgery, with a focus on addressing the disparities in access to essential and emergency surgery between HICs and LMICs. The Association of Surgeons in Training (ASiT) convened a consensus meeting in Liverpool in 2016 to explore the roles and barriers for trainees in global surgery. The paper highlights the potential role of surgical trainees in global surgery and emphasizes the importance of ethical considerations and collaboration with local surgeons in terms of on-site teachings in LMICs. It suggests the development of appropriate pathways for recognizing global surgery experience for trainees [48].

Another study by Taub et al. from 2015 discusses the challenges and safeguards associated with volunteer craniofacial surgical missions in LMICs. The organization KomedyPlast was established in 2001 to provide craniofacial surgical care to underserved populations and to educate local surgeons. Over the course of nine annual missions, KomedyPlast has provided surgical care to over 150 patients with complex craniofacial conditions in LMICs. The article emphasizes the importance of implementing safeguards to ensure safety and minimize risks during these missions [49].

Nagengast et al. performed a study which focuses on the transition of Operation Smile, an organization providing surgical care in LMICs. The aim is to deliver surgical services within the country of need rather than relying solely on international teams. A retrospective review of the Operation Smile mission database for the years 2014 to 2019 revealed that, on average, 144.8 surgical missions per year were conducted during the study period. Once these teams are in place, conducting local missions becomes an effective approach to provide specialized surgical care within a country’s own borders. By shifting to the local mission model, Operation Smile aims to enhance access to quality surgical care and build sustainable surgical capacity within the countries they serve [50].

Cook et al. describe in a study from 2015 the Alliance for Global Clinical Training. This is a consortium of US surgical departments that aims to provide continuous educational support to surgeons at the Muhimbili University of Health and Allied Sciences (MUHAS) in Tanzania. This study evaluated the effectiveness of the Alliance in meeting the educational needs of MUHAS and Muhimbili National Hospital surgeons. Anonymous surveys revealed that a multi-institutional international surgical partnership, such as the Alliance, is feasible and leads to perceived improvements in patient care and resident learning [51].

Many academic medical centers (AMCs) are establishing partnerships with teaching hospitals overseas to address these issues. The authors emphasize the importance of relationships, mutual learning, local training needs, collaboration in research, adapting to local needs, a multidisciplinary approach, and measuring outcomes for successful partnerships [52].

A review by Guntaka et al. from 2022 aimed to characterize the landscape of global academic collaborations in the field of oral and maxillofacial surgery (OMS) between HICs and LMICs. The review identified 71 articles describing 81 unique collaborations between HICs and LMICs from 1996 to 2020. The review serves as a resource for understanding current and past collaborations in global OMS, identifying areas for capacity building, and guiding future research efforts in this field [53].

#### 3.1.3. Fellowships Abroad

Shaye et al. presented in a 2018 published study a teaching concept for surgeons from LMICs. CMF trauma surgeons from different countries were recruited to develop a curriculum for treating CMF trauma in LMIC settings. They used a modified Delphi method to identify the most common patient problems related to CMF trauma and ranked them based on morbidity. The goal is to improve the care of facial injuries in low-resource settings, acknowledging the significant burden they pose and the need for specialized training in these contexts [54].

Bhandari described a fellowship for CMF surgery in a study from 2021. Residency training in oral and maxillofacial surgery, plastic surgery, or otolaryngology does not adequately cover all aspects of craniofacial surgery, making additional fellowship training necessary for fresh graduates to become independent craniofacial surgeons. In this article, the author provides a critical review of their one-year craniofacial surgery fellowship at Chang Gung Memorial Hospital in Taiwan, sponsored by the Noordhoff Craniofacial Foundation (NCF). The author describes their rotations in pediatric craniofacial surgery, orthognathic surgery, and craniofacial trauma and reconstruction, and presents a surgical log along with a critical evaluation [55].

### 3.2. Category II: Transfer of the Patient to Specialized National Clinics

Another way to care for patients with CMFs in LMICs is to transfer them nationally to a specialized hospital.

Porter et al. analyzed the number and reasons of CMF patients’ transferrals in South Africa in a study from 2013. South Africa’s healthcare system consists of a public and private sector, with the public sector catering to approximately 80% of the population. The public healthcare system is categorized into district, regional, tertiary, and quaternary hospitals, each offering different levels of care. The Division of Maxillofacial and Oral Surgery at the University of the Witwatersrand provides specialized care at Charlotte Maxeke Johannesburg Academic Hospital and a tertiary hospital in Soweto.

This investigation aimed to identify the primary cause of delayed treatment for facial fractures. Patient records from 2002, 2004, and 2006 were examined, and various factors related to timing were recorded.

The results showed that facial fractures were being treated approximately 20.4 days after occurrence, with delays occurring in both the time from fracture occurrence to presentation at the Division of Maxillofacial and Oral Surgery and from presentation to treatment. Patients were presented to the Division of Maxillofacial and Oral Surgery, on average, 10 days after the fracture. Factors such as shortages of anesthetists, operating theatre staff, and facilities for general anesthesia contributed to treatment delays.

To address the issue of delayed treatment, the study suggests increasing public awareness about the importance of early diagnosis and treatment of facial fractures through media campaigns. Overall, the study highlights the multifaceted nature of the problem and proposes targeted strategies to minimize delays in the treatment of facial fractures in the South African healthcare system [56].

Stanford-Moore et al. conducted a prospective cohort study in Rwanda to examine the impact of delays in diagnosis and treatment on outcomes of CMFs in 2022. The study included all patients with CMFs who presented to the emergency department of a referral center between 1 June and 1 October 2020. Patients follow up was carried out for a minimum of 6 months to assess outcomes.

The study found that delays in treatment were associated with a higher risk of complications. Patients who experienced a delay of ≥3 days in treatment after arrival to the hospital were 4.25 times more likely to have a complication compared to those without treatment delay. However, there was no statistically significant increase in complication rate for patients who arrived at the hospital ≥ 3 days after injury. The predominant injury observed was mandibular fracture (*n* = 28), succeeded by occurrences of zygomatic fracture, frontal bone fracture, or soft tissue injury (*n* = 8 each). Among individuals with mandibular fractures, 14 cases involved isolated mandibular fractures, while 14 others presented with associated injuries, with LeFort III fractures being the most prevalent (*n* = 3). Unfortunately, the study does not specify which of the fractures were open or closed.

Patients coming from rural settings were more likely to experience delays in presentation to the hospital. In conclusion, the study demonstrated the importance of timely diagnosis and treatment in CMF patients and delays in treatment were associated with an increased risk of complications [57].

### 3.3. Category III: Off-Label and Non-Operative Solutions

In a retrospective study we showed an off-label treatment option using a hand fixator system as an external face fixator for CMFs, providing a detailed surgical technique description. The feasibility and postoperative outcomes of the off-label surgical technique were evaluated by analyzing a cohort of 13 CMF patients treated at an NGO hospital in Sierra Leone between 2015 and 2019. All midface fractures were managed accordingly. Detailed classification was difficult to impossible due to the lack of CT scans and was conducted solely based on two-view X-rays and clinical examination during the operation. The fixators were applied in a “bottom-up” fashion. If the mandible was fractured, it was addressed first, followed by closure and fixation of any potential hard palate fracture. Reduction was performed under direct visualization and through temporary occlusion. If the midface could be dislocated ventrally from the head, a type of Le Fort 2/3 fracture was assumed, and the fixator was supplemented with supraorbital fixation. The authors conclude that the usage of a hand fixator system as an external face fixator was feasible. Notably, dynamic occlusion could be maintained during the 6-week healing period, enabling normal oral functions such as eating, drinking, and dental hygiene. This approach offers improved outcomes and enhanced quality of life for patients by maintaining dynamic occlusion and allowing normal oral functions during the healing period [23].

Another research group led by Hihara et al. presented a similar therapeutic approach in their study published in 2019. This case report documents the treatment of a patient with CMF injuries utilizing an Ilizarov fixator. Complicated by a concurrent vascular injury, the patient experienced hemodynamic shock, necessitating the postponement of initial CMF therapy in favor of addressing the vascular injury through embolization. The Ilizarov fixator is an orthopedic external fixation device widely used in limb reconstruction and lengthening procedures. It was developed by Dr. Gavriil Abramovich Ilizarov, a Russian orthopedic surgeon, in the 1950s [58].

The fixator was left in place for a total of 8 weeks, and occlusion was successfully restored [59].

In their 2014 study, Beogo et al. conduct a retrospective review assessing the efficacy and complications associated with wire osteosynthesis as an internal fixation method for facial fractures in a university teaching hospital in Burkina Faso. The records of 227 patients treated with wire internal fixation at the Department of Stomatology and Maxillofacial Surgery of Centre Hospitalier Universitaire (CHU) Sourô Sanou between 2006 and 2010 were analyzed. The study found a satisfactory treatment outcome in 91.2% of the patients. Considering the limited resources, wire internal fixation may be a viable alternative for the surgical treatment of noncomminuted facial fractures without bone substance loss [60].

In their study, Cienfuegos et al. analyzed 45 patients between 4 and 56 years of age with palatal fractures. The fractures were treated using medium- or high-profile locking plates placed over the palatal mucosa as an external fixator, along with treatment for other associated facial fractures. At 12 weeks, the plates and screws used for palate fixation were removed once computed tomography scans confirmed fracture healing. The results showed that all palatal fractures had healed within 12 weeks [61].

## 4. Discussion

The elevated prevalence of CMF injuries, coupled with a scarcity of infrastructure, resources, and training, presents a significant hurdle in treating these severe cases in LMICs.

Broadly, the identified solutions can be categorized into three primary areas:

Facilitating Education and Training: Prioritizing the education and training of surgeons in LMICs to enhance their capabilities in managing CMF cases.

Patient Transfer to Specialized Centers: Transferring affected patients to specialized national centers equipped to provide the necessary therapeutic interventions.

Exploring Off-label Solutions: Leveraging off-label solutions to optimize available resources, ensuring the delivery of the best possible care to the patients.

Overall, awareness programs for the local population to consult specific treatment in cases of CMFs are inevitable for prompt and appropriate treatment to minimize the long-term consequences of facial trauma and promote optimal recovery.

The proposed solutions for improving the therapy of CMF patients in LMICs, as described in the literature, are diverse, as are the challenges. There is a lack of well-trained surgeons, osteosynthesis materials and other consumables, as well as urgently needed infrastructure. The long-term goal should be to align the therapy of CMFs in LMICs with that in HICs [62]. Achieving this goal necessitates a synergistic approach, involving both education and training initiatives and the augmentation of local resources [63,64,65,66]. However, numerous challenges need to be overcome before this goal can be accomplished.

In the interim, the emphasis should be on concurrently fostering lasting changes in the current scenario and optimizing opportunities for the immediate enhancement of CMF therapy [67].

Improving education and training as well as passing on operational skills appears to be the most promising approach in the medium and long term. With today’s possibilities of digital teaching through lectures, guidelines, and surgical videos, the on-site expertise in LMICs can thus be significantly increased [68]. Especially the theoretical aspects, which are essential for the therapy of CMFs, can be adequately taught in this way. The included studies dealing with digital teaching of surgical techniques for oral and maxillofacial surgeons concluded that this teaching method can improve skills. Studies that analyzed online knowledge sharing in other specialties also had positive outcomes [69,70,71,72,73,74]. A disadvantage of this digital approach is the lack of hands-on courses. One way of passing on these technical skills online is to set up hybrid courses. Another possibility is to build up a global network of experts from HICs and local surgeons from LMICs.

Through this approach, fellowships can be offered in specialized clinics, facilitating the transfer of the required surgical skills [75]. A disadvantage of this approach is the necessary financing of the fellow’s stay. In addition, during this time he is absent as a surgeon in his home country, which usually already has a significant shortage of surgeons. The last possibility to share the needed skills is to teach the local surgeons on site, for example during a humanitarian mission. For this purpose, projects exist that discuss complex cases online in advance and then provide care together during the mission.

However, transferring essential surgical skills to local surgeons, regardless of which of the aforementioned techniques, has the disadvantage that the improvement of local patients will not be visible for several years. The big advantage, however, is that the well-trained local surgeons can in turn train younger colleagues, making the system more and more independent of HICs.

The transfer of CMF patients has the advantage that they are likely to receive state-of-the-art therapy at a nationally specialized clinic. However, there are two disadvantages to this approach: Firstly, in these countries, patient transfer is often very time-consuming due to transportation infrastructure, and this delay can lead to a worse outcome when compared to patients who received immediate therapy [56,57]. Furthermore, the transfer of critically injured CMF patients is often not feasible [76,77]. Secondly, specialized cranio-maxillofacial centers are scarce in most LMICs, making this alternative unavailable in those regions, and requiring the patient to be referred for an alternative treatment [7].

In this context, off-label techniques have two major advantages: First, the patient can receive prompt and effective on-site treatment, and second, resources commonly found in remote rural hospitals are utilized, despite potential limitations compared to state-of-the-art therapies in HICs. Second, Employing these techniques effectively maximizes the utilization of local resources and holds the potential to enhance treatment outcomes, particularly in regions with restricted access to specialized medical facilities or advanced medical equipment [23,59,60,61,78]. However, a disadvantage is that mastering these techniques requires training, and the outcomes may be limited compared to state-of-the-art therapies in HICs. Therefore, the implementation of these alternative off-label techniques appears to be reserved for situations in which any other form of adequate surgical care is impossible.

In conclusion, a multi-faceted approach that combines education and training of local surgeons, access to digital learning platforms, and utilization of off-label solutions is necessary to address the challenges of CMF trauma in LMICs. By empowering local surgeons and providing them with the necessary skills and knowledge, sustainable surgical care can be achieved in these regions. Collaboration between international organizations, local healthcare providers, and educational institutions is vital in implementing these solutions effectively and achieving meaningful impact in improving patient outcomes in LMICs.

## Figures and Tables

**Figure 1 jcm-13-02437-f001:**
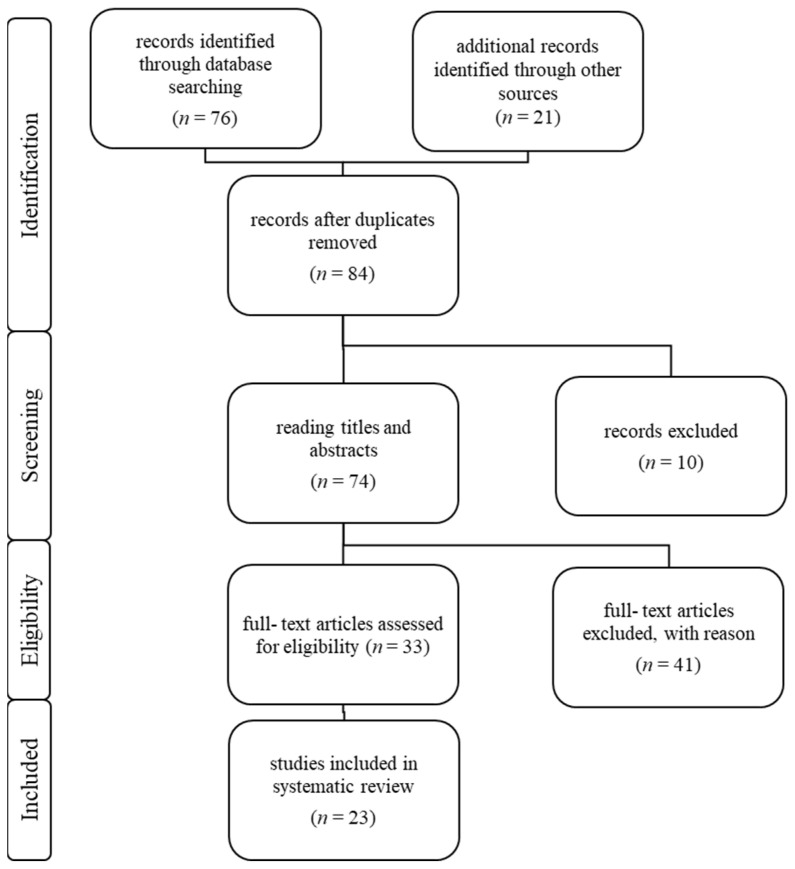
Overview of the study selection following recommendations of the PRISMA-Statement.

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
