# Peer review of "Enhancing Cranio-Maxillofacial Fracture Care in Low- and Middle-Income Countries: A Systematic Review"

_jcm, 2024, doi:10.3390/jcm13082437_

Round 1
Reviewer 1 Report
Comments and Suggestions for Authors
The article is well written. However, I feel that its incomplete when we are talking about low to middle income areas, just educating the surgeons and providing better healthcare won't suffice.
Educating the masses, making the common man aware of the need for seeking treatment when they are affected by maxillofacial trauma is also prime important.
The population from rural or remote areas, even when they are aware that a hand fracture needs treatment, they lack the awareness that a trauma to the face can cripple them for life if untreated. The amounts to a high percentage of post traumatic deformity.
Kindly add the facts related to the same.
The line that quotes reference [15,16] needs to be rewritten for clarity. The term 'further injurie and restrictions' can be reframed as post traumatic deformity.
Also, when referring to LMIC, the article is biased towards few regions and deficient in covering a lot of grave concerns of vast majority of other LMIC countries.
Author Response
Dear Reviewer,
Thank you very much for your assistance and comments. We have revised the manuscript accordingly. The revised manuscript and response letter are attached.
With kind regards, The Authors

Reviewer 2 Report
Comments and Suggestions for Authors
The study addresses the enhancement of craniomaxillofacial fracture care in LMICs. I am very glad to hear that also orthopedic surgeons know well about maxillofacial traumatology. Therefore, it does not seem to be necessary to include maxillofacial surgeons in the list of authors. You provided a very interesting work about maxillofacial education and care in LMICs. I would suggest the following amendments:
Please specify what are the common CMFs in detail.
Mandibular fractures represent a large proportion of facial injuries. Except for fractures in the temporomandibular joint area, they require ORIF in most cases. Post-treatment malocclusion and mouth opening reduction are the main complications. I would go into it in one sentence in the introduction with corresponding citation.
Another main complication of CMFs is reduced aesthetic outcome due to healing in malposition (e.g. cheek bone fracture) and loss of vision. I would also mention that in the introduction.
3.3 ..."The study found that delays in treatment were associated with a higher risk of complications. Patients who experienced a delay of ≥3 days in treatment after arrival to the hospital were 4.25 times more likely to have a complication compared to those without treatment delay. However, there was no statistically significant increase in complication rate for patients who arrived at the hospital ≥3 days after injury."
I recommend discussing what kind of fractures were included in this study. Fractures of the mandibular body are mostly open fractures which require urgent treatment whereas fractures of the midface are commonly treated when the accompanying swelling has disappeared some days after the trauma. A certain treatment delay therefore makes sense in some cases, and it does not make sense to include all kinds of facial fractures in one study from my opinion.
3.3 ..."dynamic occlusion" says tooth contact during movement. I think you mean "habitual occlusion" or "habitual intercuspidation" (teeth fit together) or maximum mouth opening respectively simple opening movements.
What certain kind of fractures were treated with this external face fixator? "CMF" is a very broad term...
I recommend mentioning that the use of the external face fixator and the Ilizarov fixator is required for cases in which standard maxillofacial treatment is definitely not available.
The technique of Cienfuegos is used in European units too, from my opinion it is the only technique to treat sagittal palate fractures. Please discuss.
Overall the study is well written and interesting. I recommend a resubmission after overthinking my suggestions.
Author Response

(The authors gave the same response as above.)

Reviewer 3 Report
Comments and Suggestions for Authors
1. In Material and Methods the authors must specify which combinations of terms have been used for the paper.
2. In the same section there is written: "Forty-one articles were excluded with reason". These reasons need to be explained.
Author Response

(The authors gave the same response as above.)

Reviewer 4 Report
Comments and Suggestions for Authors
Dear authors!
You touched on a very sensitive and broad topic!
The conclusions you received really deserve the close attention of the professional community.
I would like to clarify your opinion - do you think it might be worth touching on the topic of specialized literature in the form of a treatment guide? These guidelines would outline specific steps and time frames for various entities.
I think that your article is very useful for the development of traumatology in maxillofacial surgery and can be an excellent start to a large project!
Author Response

(The authors gave the same response as above.)
